# Pathological Features of Echovirus-11-Associated Brain Damage in Mice Based on RNA-Seq Analysis

**DOI:** 10.3390/v13122477

**Published:** 2021-12-10

**Authors:** Guoyan Zhang, Jichen Li, Qiang Sun, Keyi Zhang, Wenbo Xu, Yong Zhang, Guizhen Wu

**Affiliations:** 1WHO WPRO Regional Polio Reference Laboratory, National Health Commission Key Laboratory for Biosafety, National Health Commission Key Laboratory for Medical Virology, National Institute for Viral Disease Control and Prevention, Chinese Center for Disease Control and Prevention, 155 Changbai Road, Beijing 102206, China; zhanggy9407@163.com (G.Z.); jichenli666@163.com (J.L.); sunqiang@ivdc.chinacdc.cn (Q.S.); zhangky17@163.com (K.Z.); wenbo_xu1@aliyun.com (W.X.); 2Biosafety Level-3 Laboratory, National Health Commission Key Laboratory for Biosafety, National Institute for Viral Disease Control and Prevention, Chinese Center for Disease Control and Prevention, 155 Changbai Road, Beijing 102206, China; 3Department of Medical Microbiology, Weifang Medical University, Weifang 261053, China; 4Center for Biosafety Mega-Science, Chinese Academy of Sciences, Wuhan 430071, China

**Keywords:** Echovirus 11, IFNAR−/− mice, aseptic meningitis, RNA-Seq, brain, human glioma cells (U251)

## Abstract

Echovirus 11 (E11) is a neurotropic virus that occasionally causes fatal neurological diseases in infected children. However, the molecular mechanism underlying the disease and pathological spectrum of E11 infection remains unclear. Therefore, we modelled E11 infection in 2-day-old type I interferon receptor knockout (IFNAR−/−) mice, which are susceptible to enteroviruses, with E11, and identified symptoms consistent with the clinical signs observed in human cases. All organs of infected suckling mice were found to show viral replication and pathological changes; the muscle tissue showed the highest viral replication, whereas the brain and muscle tissues showed the most obvious pathological changes. Brain tissues showed oedema and a large number of dead nerve cells; RNA-Seq analysis of the brain and hindlimb muscle tissues revealed differentially expressed genes to be abundantly enriched in immune response-related pathways, with changes in the Guanylate-binding protein (GBP) and MHC class genes, causing aseptic meningitis-related symptoms. Furthermore, human glioma U251 cell was identified as sensitive target cells for E11 infection. Overall, these results provide new insights into the pathogenesis and progress of aseptic meningitis caused by E11.

## 1. Introduction

Enteroviruses can cause many human diseases, including acute meningitis (AM), acute flaccid paralysis, conjunctivitis, myocarditis, and pericarditis, most of which are mild and self-limited [1]. However, certain enteroviruses may cause severe or even fatal neurological diseases. In recent years, Echovirus 11 (E11) has caused increasing outbreaks, especially in young children. E11 is reported to be more likely to cause neonatal infections compared to coxsackievirus B5 (CVB5). In 2018, E11 emerged as the predominant enteroviral strain in Taiwan; confirmed cases were found across the province, of which 35 were neonatal infections, and 7 of 8 severe cases died [2]. In May 2019, in Guangzhou City, Guangdong Province, an in-hospital neonatal infection incident was caused by nosocomial infection of E11, wherein five infants with neonatal pneumonia and other underlying diseases died [3]. In July 2019, an outbreak of E11 infection occurred in newborns in Enshi City, Hubei Province, and seven neonates died. E11 has also been frequently identified as the causative agent of outbreaks in countries such as India, Thailand, Japan, and Israel [4,5,6]. E11 belongs to group B enteroviruses and is a significant pathogen of aseptic meningitis in young children. The dominant manifestation of E11 infection is respiratory illness with symptoms including cough and rhinitis. The systemic symptoms of AM include high fever lasting four to five days, and typical symptoms include persistent headache and vomiting, suggesting central nervous system involvement but not altered sensorial or focal neurological findings [1]; these symptoms are often severe and even fatal. Compared to children infected with CVB5, children infected with E11 show more vital inflammatory activity, resulting in longer hospitalisation [2].

Viruses are strictly parasitic microorganisms. To complete their replication and infection cycle, viruses must rely on host factors [7]. In general, once a virus enters the cell, the host innate immune response produces many factors that play a key role in inhibiting viral replication. If these are insufficient, viral replication continues and causes disease development [8]. Therefore, host factors play a vital role in regulating the viral replication and host antiviral activity. Currently, there have been few studies examining the pathogenic mechanism of E11 infection. Thus, the gene expression profile of the host response to E11 remains unclear, and there is an urgent need to reveal its pathogenesis at a molecular level; therefore, additional research on the pathogenic mechanism of E11 is essential [9].

In this study, we used an animal model to better understand the neuropathology and neurovirulence of E11 infection and its molecular mechanism. For simulating the clinical symptoms of E11 infection in neonatal children, we injected clinically isolated E11 strains (HuB2019-16 isolated during an outbreak of AM in Hubei province in 2019) into 2-day-old type I interferon-receptor knockout (*IFNAR*−/−) mice. We studied the corresponding pathological changes and performed RNA-Seq to determine a comprehensive analysis of the host gene changes caused by E11infections.

## 2. Methods

### 2.1. Ethics Statement

Specific-pathogen-free (SPF) immunocompromised *IFNAR*−/− mice were used to develop an animal model. The Ethics Review Committee of National Institute for Viral Disease Control and Prevention, Chinese Center for Disease Control and Prevention, approved the animal experiments conducted (permission no. 20201022059).

### 2.2. Mice, Cells, and Viruses

Specific pathogen free immunocompromised *IFNAR*−/− mice were purchased from the Institute of Medical Laboratory Animals, Chinese Academy of Medical Sciences (permission no. SCXK Jing 2019-0014). All mice were kept at the life science laboratory animal facility and were housed in independently ventilated cages. The cage was lined with clean sawdust supplemented with cotton for nesting. Mice were mated to obtain pups for animal experiments.

Human rhabdomyosarcoma (RD) cells (ATCC) were cultured in Dulbecco’s modified Eagle’s medium (DMEM) (Gibco, the New York City, NY, USA) supplemented with 10% foetal bovine serum (Gibco, USA), 100 IU penicillin (Hyclone, Logan, UT, USA), and 100 μg streptomycin (Hyclone, Logan, UT, USA) per millilitre at 37 °C in the presence of 5% CO_2_. Human neuroblastoma cells (SK-N-SH) and human glioma cells (U251) were purchased from the National Infrastructure of Cell Line Resource and cultured in a 1:1 mixture of DMEM and F-12 (Gibco, the New York City, NY, USA) supplemented with 10% FBS. The E11 virus strain (HuB2019-6) used for the challenge was previously isolated from a throat swab sample of a girl who had died as a newborn during an outbreak in Enshi, Hubei, in 2019. This virus was propagated and titrated in RD cells (1 × 10^8^ TCID_50_/mL) and stored at −80 °C. The titres of E11 were expressed in TCID_50_ following the Reed and Muench method. After thawing at 4 °C, the virus suspension was brought to room temperature (RT) for the animal challenge experiment.

### 2.3. Animal Infection Experiments

Preliminarily, we determine the sensitivity of 2-day-old *IFNAR*−/− mice to E11 infection. Virus injection was performed using an ultra-fine insulin syringe (Beckton, Dickinson, and Company, Franklin Lakes, NJ, USA), within an animal biological safety cabinet (ABSL II) to reduce animal suffering. Through the pilot experiment, we found that the symptoms of mice were most obvious by intracranially injection, and intracranial injection can better simulate the clinical symptoms of infant infection. Therefore, the 2-day-old *IFNAR*−/− mice (n = 16 per group) were intracranially (i.c.) injected with 40 µL of the E11 suspension (virus titre, 10^8^ TCID_50_/mL, diluted 10-fold to 10^7^ TCID_50_/mL); mice in the control group were injected with 40 µL of MEM as MOCK-infected mice. Body weight trends, survival rate, clinical scores, and illness were recorded daily until six days post-inoculation. The grade of clinical disease was scored as 0, healthy; 1, lethargy or inactivity; 2, wasting or hind limb weakness; 3, single hind limb paralysis; 4, double hind limb paralysis; and 5, death [10]. When the neonatal mice became moribund, their brains, hearts, lungs, spleens, intestines, and contralateral hind limb skeletal muscles were aseptically removed after anaesthetization and stored at −80 °C for subsequent experiments.

### 2.4. Virus Titration

The brain, hind limb muscle, heart, lung, and small intestine tissues were obtained at 2, 4, and 5 days post infection (dpi). The samples were washed with sterile PBS to remove excess blood, and 700 µL of PBS containing 1% penicillin and streptomycin was added to each tissue. The supernatant was then ground with a Scientz high-throughput tissue grinder (Ningbo SCIENTZ Biological Technology Co., Ltd., Ningbo, China), then disrupted by three freeze–thaw cycles, and then take the supernatant after centrifugation. Viral titration was measured from sets of 96-well plates. The viral titres from 10-fold serial dilutions of each sample were evaluated using a 50% TCID_50_ assay on RD cells. The measurement was repeated 3 times at each time point (n = 5). U251 and SK-N-SH cells were infected with a multiplicity of infection (MOI) at 10, and RD cell was infected with a MOI at 0.005. Infection medium was washed three times with PBS at 2 h post infection (hpi) to reduce background. Cell culture lysates were collected at four time points (12 hpi, 24 hpi, 48 hpi, and 72 hpi) after infection, and the viral titres were measured using endpoint dilutions for growth in RD cells. The measurement was repeated 3 times at each time point.

### 2.5. Histology of E11-Infected Mice

E11-infected brain tissues, hindlimb muscles, heart, lung, and small intestine tissues were fixed in neutral buffered formalin for 3 to 5 days. They were trimmed into standard cross-sections, routinely processed, embedded in paraffin, and sectioned. Sections (4 µm) were mounted on 3-aminopropyltriethoxysilane-coated slides and stained with haematoxylin and eosin (HE) for light microscopic evaluation. The AxioCam MRc5 (Carl Zeiss) system was used to obtain the results at magnifications of ×200 and ×400.

### 2.6. Transcriptome Sequencing

The collected brain tissues and hindlimb muscle tissues were lysed using TRIzol and ground. They were then sent to Novogene Technology Co., Ltd. (Beijing, China), for transcriptome sequencing on an Illumina HiSeq TM4000 sequencer.

### 2.7. RNA-Seq Data Analysis

MapSplice software was used to align the sequences for determining the differentially expressed genes. For expression-based analysis, an online bioinformatics tool was used to convert numerical data into bar graphs (http://www.bioinformatics.com.cn/static/others/daohang.html) (accessed on 5 November 2021). The online server DAVID (https://david.ncifcrf.gov/tools.jsp) (accessed on 5 November 2021) was used to determine gene ontology (GO) and for the functional enrichment analysis of differentially expressed genes and Kyoto Encyclopaedia of Genes and Genomes (KEGG) pathways.

### 2.8. Immunofluorescence Analysis of Brain Tissues

Mice with a clinical score ≥ 4 were anaesthetised using ether and sacrificed. Their brain tissues were fixed in 10% PFA, dehydrated, embedded in paraffin, cut into 4 µm slices, and attached to glass slides. For staining, the sections were fixed in 4% PFA for 15 min, washed with PBS, permeabilised using 0.5% TritonX-100 in PBS, and then blocked with 2% bovine serum albumin for 30 min. The sections were incubated overnight at 4 °C with polyclonal mouse anti-dsRNA J2 antibody (1:100 dilution, Scicons, Hungary), followed by incubation with fluorescence-tagged Goat Anti-Mouse IgG antibody, Dylight 680 (1:500 dilution, ABclonal, Wuhan, China) at RT for 1 h.

### 2.9. Bright-Field Microscopy

The morphology of E11-infected and Mock-infected RD, SK-N-SH, and U251 cells was observed for cytopathic effect (CPE) at 24 h using a bright field Leica inverted microscope (Leica Microsystems, Weztlar, Germany). The brightness and contrast of all images were adjusted using Image-Pro Plus 6.0.

### 2.10. Immunofluorescence Analysis of Cells

The cells were washed with PBS, fixed with 4% formaldehyde for 15 min, washed three times with PBS, and permeabilised with 0.2% Triton X-100 for 20 min at RT. After three washes with PBS, the tissues were blocked with 5% bovine serum albumin for 30 min and subsequently incubated with a specific primary rabbit polyclonal antibody targeting VP1 of Echovirus 11 (Reacts with the Echovirus 4, 6, 9, 11, 30 and 34) (1:200 dilution, ABclonal, Wuhan, China) overnight at 4 °C. The cells were then washed and incubated with FITC-conjugated goat anti-rabbit IgG (1:500 dilution) (Abbkine Scientific Co., Ltd., Wuhan, China) or FITC-conjugated goat anti-mouse IgG (1:500 dilution) (Abbkine Scientific Co., Ltd., Wuhan, China) secondary antibodies for 1 h at RT. Finally, after nuclei staining with 4,6-diamidino-2-phenylindole solution (DAPI), the cells were washed and mounted for observation of immunofluorescence using a Leica inverted microscope (Leica Microsystems, Weztlar, Germany).

## 3. Results

### 3.1. E11 Challenge in Neonatal IFNAR−/− Mice

In a preliminary experiment, we demonstrated that all 2-day-old *IFNAR*−/− mice were susceptible to i.c. infection with E11(HuB2019-16) at a dose of 10^7^ TCID_50_/mL. Furthermore, the clinical symptoms produced by the inoculation of 10-fold TCID_50_-diluted E11 in mice can simulate the symptoms of human infant infections. The neonatal mice began to show clinical signs such as weight loss, listlessness, ataxia, paralysis of one or both hind limbs after challenge from 1 to 5 dpi, and all mice died by 5 dpi (Figure 1). The control group did not show any clinical manifestations or symptoms of weight loss.

### 3.2. Dynamic Monitoring of Viral Loads in Different Organs

The TCID_50_ method was used to detect changes in the E11 viral loads in the brain, hind limb muscle, heart, lung, small intestine, and spleen tissues at 2, 4, and 5 dpi after being challenged with E11 via i.c. inoculation in mice. The results showed that the viral loads differed significantly among tissues (Figure 2), and the viral loads in each organ were low at 2 dpi (Figure 2). The viral load in the muscle was the highest. All the viral loads were greater than 1 × 10^4^ TCID_50_/mL but were lower than that in the middle and late stages of infection. By the middle and late stages of the infection, the viral load of all tissues, including the brain, was increased to varying degrees, indicating that virus replication had progressed in the brain tissue (Figure 2). Furthermore, in E11-infected mice, the highest mean titres were observed in skeletal muscles at all time points, indicating that the E11 virus has a clear tendency to replicate in muscle tissues. The virus replicated rapidly in the hind limb muscles, causing muscle paralysis and hind limb paralysis. The virus titre was lowest in the spleen.

### 3.3. Pathological Observations

Generally, viral proliferation in tissues can induce a host’s inflammatory response, even in the central nervous system (CNS) [11]. Thus, the evidence of viral infection included hyperaemia, oedema, inflammatory cell infiltration, and tissue necrosis [12]. Two-day-old neonatal mice were challenged with the E11 HuB2019-16 strain (virus titre, 10^8^ TCID50/mL, diluted 10-fold to 10^7^ TCID50/mL) via the i.c. route, and animals were sacrifice when the clinical scores reached four; the tissues were then surgically removed, processed for histological sectioning, and subjected to HE staining. A pathological examination showed that many nerve cells, especially glial cells, were significantly reduced in E11-infected mice. Furthermore, the brain tissues displayed softening of the cribriform foci, with many oedematous neurones, while the cytoplasm contained vacuoles (Figure 3A (1–6)). The effective replication of E11 leads to the appearance of aseptic meningitis-related symptoms such as fever and cerebral oedema. The hind limb muscles were severely affected and showed muscle fibre necrosis and muscle bundle rupture (Figure 3B (1–6)). In the late stage of infection, severe viral myocarditis was observed with myocardial lysis, degeneration, and capillary leakage. Large numbers of diffuse lymphocytic and mononuclear cell infiltrates were evident and were associated with myocardial rupture (Figure 3C (1–6)). The lung tissues showed a significantly widened alveolar space, vasodilatation, hyperaemia, pulmonary fibrosis, and interstitial oedema with large numbers of inflammatory cell infiltrates (Figure 3D (1–6)). Enteritis characterised by the vacuolar degeneration of the small intestinal villi was observed, accompanied by massive inflammatory cell infiltration (Figure 3E (1–6)). Significant enlargement and translucency of hepatocytes were observed, showing loose cytoplasm, and hepatocytes changed from polygonal to spherical, with inflammatory cell infiltration, mainly characterized by lymphocyte infiltration in the portal area and vascular subendothelial infiltration of eosinophils (Figure 3F (1–6)). Glomeruli showed increased cellularity with proliferation and swelling of endothelial and mesangial cells (Figure 3G (1–6)). However, no significant pathological changes were observed in the spleen. There were no noticeable pathological changes in the tissues of the negative control group, indicating that E11 infection can cause severe pathology, tissue lesions, and inflammatory reactions.

### 3.4. Transcriptional Changes in E11-Infected Brain and Muscle Tissues

To investigate the transcriptome responses to E11 infection in the brain and muscle of neonatal mice, differentially expressed transcripts were screened based on the criterion of log2(fold change) ≥ |±1.2| and *p*-value < 0.05. In total, 375 differentially expressed genes were significantly altered in the brain, with 281 upregulated and 94 downregulated genes. However, a comparison of the E11-infected and control groups in the muscle identified 1181 genes (816 upregulated and 365 downregulated) that were significantly differentially expressed. The numbers of upregulated and downregulated genes in each group are shown in Figure 4A. A Venn diagram was constructed to identify the common differentially expressed genes in the brain and muscle groups, and 161 differentially expressed genes were overlapping in each group (Figure 4B). We then performed GO and KEGG enrichment analyses of these differentially expressed genes to compare the host responses in the brain and muscle. Genes related to the immune system process, inflammatory response, defence response to viruses, immune response, cytokine–cytokine receptor interaction, the chemokine signalling pathway, innate immune response, neuronal cell body, the apoptotic process, the Toll-like receptor signalling pathway, the negative regulation of viral genome replication, the TNF signalling pathway, the NF-kappa B signalling pathway, and T cell receptor binding were activated in the brain sites (Figure 4C). In the muscle, the differential expression of genes involved in the inflammatory response, immune response, innate immune response, defence response to a virus, response to a virus, MAPK cascade, complement activation (classical pathway), complement activation, the negative regulation of neuron projection development, and astrocyte cell migration was found at the “Biological process” level, whereas genes involved in neuroactive ligand–receptor interaction, complement, and coagulation cascades, the Toll-like receptor signalling pathway, and the NOD-like receptor signalling pathway were activated at the “KEGG” level (Figure 4D). On one hand, activation of the immune response, both in the brain and muscle, could provide mice with effective immune protection against E11 infection. On the other hand, this might induce excessive cytokine production, consistent with “cytokine storm” in patients at severe and critical stages of AM [13,14]. As shown in Figure 4E,F, these immune processes in both the brain and muscle included complement transcription factors (e.g., C1ra, C3ar1, and C5ar1), regulators of cellular response to interferon γ (e.g., Gbp2, Gbp3, Gbp4, Gbp5, Gbp6, Gbp7, Gbp9, Gbp10), immune responses (e.g., H2-Aa, H2-Eb1, H2-K1, H2-Q4, H2-Q6, Tgtp1, and Serpina3g), and defence response factors (e.g., Irgm1 and Tgtp2) [15]. We observed that these common differentially expressed genes were upregulated by 1.27- to 12.25-fold (Figure 4E,F).

### 3.5. E11 Replication in the Mouse Brain and in U251 Cells

Considering that the brain is one of the most significant target organs for E11 infection, we assessed the infection and replication of E11 in the mouse brain. Immunofluorescence staining revealed that the viral dsRNA was detected in the mouse brain region (Figure 5A). To determine the most significant target cells for E11 infection in the brain, we evaluated E11 infection and replication in U251 human glioma cells and in SK-N-SH human neuroblastoma cells using the sensitive RD cells as the positive control. Virus titres were measured four times after inoculation by infecting nerve cell lines at a MOI of 10 and RD cell lines at a MOI of 0.005. However, viral titration and the observation of CPE indicated that E11 replicated only in U251 cells (Figure 5B,C). VP1 is one of the initial proteins expressed after picornavirus infection before capsid assembly [16]. Therefore, we also examined the VP1 capsid protein of E11 during infection of U251 cells at 24 h post-infection (hpi) using immunofluorescence analysis. VP1 protein was detected in U251 cells following infection with E11, consistent with the viral replication experiments (Figure 5D).

## 4. Discussion

E11 is one of the most frequently isolated enteroviruses causing AM worldwide in recent years [3]. *IFNAR*−/− mice are sensitive to enteroviruses [17]. In this study, we used RNA-Seq technology to obtain information about the pathogenic mechanisms of AM induced by E11 infection. Newborn mice (≤3 days) are susceptible to E11 infection through craniocerebral injection [18]. A dose of 40 µL of E11 suspension (virus titre, 10^8^ TCID50/mL, diluted 10-fold to 10^7^ TCID50/mL) was administered to 2-day-old mice. On the first day after challenge, mice began to show symptoms such as decreased activity, reduced weight gain, and quadriplegia; all mice died on the fifth day after infection. In this study, the highest viral load was found in the skeletal muscle (10^7.7^ TCID_50_/mL), consistent with the results of other enterovirus infections [10]. Despite the relatively low viral load in the brain (10^5.25^ TCID_50_/mL), HE staining in mice showed severe pathological changes in the brain, including inflammatory cell infiltration, massive neuronal death, cerebral congestion, and oedema. These results indicate that our selected mouse model and challenge method can effectively reproduce the E11 infections observed in human infants.

EV infections can induce pro-inflammatory cytokine expression as part of the innate immune response and immune regulation to eliminate viruses [10,19]. RNA-Seq was used to investigate differences in gene expression at a genome-wide level. Compared with other methods, RNA-Seq has advantages of more accurate quantification, higher repeatability, wider detection range, and more reliable analysis [20]. RNA-Seq analysis using a Venn diagram enabled the identification of 161 common genes whose expression was altered changed after E11 infection. Enriched GO terms such as immune system process, innate immune response, defence response to a virus, response to a virus, immune response, inflammatory response, cellular response to interferon-gamma, cell surface, and defence response, as well as KEGG pathways such as cytokine–cytokine receptor interaction were considered to be closely associated with the pathogenic mechanism of E11. These key pathways may be involved in the progression of AM. In particular, the large GTPase guanylate binding protein family members (GBPs) were found to be widely expressed. GBPs are a family of 65–73 kDa GTPases that were first isolated as highly expressed proteins in murine and human cells stimulated with IFNγ [21,22,23]. The DEGs in our study were primarily enriched in the innate immune response and immune response and were mainly induced by T cells, NK cells, and NKT cells, which express type II interferon IFNγ [24]. Possibly owing to the molecular differences between IFN-α/β and IFN-γ receptors, GBP could be strongly induced by IFN-γ. Murine and human GBPs share a high degree of homology. GBPs function in various pathways, including those involved in cellular proliferation, angiogenesis, and immunity [25,26]. Human GBP1 (hGBP1) has been shown to control infections caused by vesicular stomatitis virus (VSV) and encephalomyocarditis virus (EMCV) when overexpressed in cell lines [27]. More recently, human GBP5 was found to restrict HIV-1 and other retroviruses by interfering with viral envelope glycoprotein processing [28]. Lu et al. found that hGBP-1 inhibits Coxsackievirus B3 replication in HeLa cells with a high level of hGBP-1 expression [17]. Studies have also suggested that the GBP family may clear pathogens via the inflammasome [29,30]. We thus speculate that GBPs are required for the inflammasome-dependent clearance of E11, and that the inflammatory response induced by GBP may contribute to the clinical manifestations of E11 aseptic meningitis. However, further studies on the antiviral mechanism of hGBP-1 are necessary for clarification.

We also found the high-level transcription of the H2 gene family. The classical class I histocompatibility molecules in mice, including H-2K, H-2D, and H-2L, are 45 kDa membrane bound glycoproteins [31]. The K, D, and L class I molecules function in the immune recognition of viral cell-surface antigens as well as allogeneic cells by cytolytic T lymphocytes [31]. This obligatory self-recognition is termed as H-2 restriction [32]. MHC class I molecules are recognised by CD8+ cells. T cells play a crucial role in eliminating human viral infections, and CD8+ cells play an antiviral effect mainly by releasing cytokines [33]. CD8^+^ T cells can participate in the virus clearance process after activation and are essential effector cells. Simultaneously, there is a significant correlation between the modulation of the cellular immune response and the prognosis of aseptic meningitis [34,35,36]. Virus infection of the CNS induces a localised expression of pro-inflammatory factors that precede and accompany the activation and recruitment of immune cells into the CNS [37]. During the acute disease phase, infiltrating virus-specific CD8^+^ T cells control viral replication by two different effector mechanisms: IFN-γ secretion controls viral replication in oligodendrocytes, whereas perforin-dependent mechanisms promote viral clearance from astrocytes and microglia [37,38]. Except for neuronal cells, almost all nucleated cells have MHC molecules and can present antigens to T cells. This may be related to the fact that E11 replicates only in U251 cells. We also found that complement-related genes are transcribed at high levels; thus, the complement cascade, which is the terminal effector mechanism of antibody immune responses, is also involved in the pathological process of E11 replication in mice. Using immunofluorescence analysis, we found that E11 replicated effectively in U251 glial cells and that the antigen/RNA of E11 virus was mainly expressed in glial cells in the brain, thus confirming the neurotoxicity of the E11 virus. Glial cells of the CNS are postulated to function as immune accessory cells, which may regulate immune reactivity within the CNS and activate or inhibit T cell responses [39]. The capacity of glial cells to promote or inhibit immune reactivity within the CNS is considered a potentially important element in the development and progression of autoimmune responses within the CNS [37]. Studies of resected adult cerebral tissue have also indicated the expression of major histocompatibility complex (MHC) class II antigens in microglia in situ [39].

In summary, this study provides a preliminary landscape of the transcriptome profiles of *IFNAR*−/−mice infected with E11. These data pave the way for future studies on the molecular mechanisms underlying the altered neurological symptoms induced by E11. A standard limitation of all animal model experiments is that they cannot completely reflect the state in humans. However, the animal model in our study provides a possible direction for future research on the mechanism of E11 infection. In the future, the influence of several genes on the replication of E11 in the brain or a certain drug that would treat human infants infected with E11 will be our study focus. Several important genes have been selected for meticulous host factor studies by RNA-Seq data. By overexpressing genes or knocking out the genes of the GBPs family or genes related to mice H2 family, we want to explore the specific processes that affect E11 infection. In addition, the interaction between host genes and viral genes would be a research emphasis, the influence of host genes on the expression of different viral proteins shall be our next research content. Overall, these results provide new insights into the pathogenesis and progress of aseptic meningitis caused by E11 and offer a new strategy for the in-depth study of the pathogenic mechanism of E11 and the development of vaccines.

## Figures and Tables

**Figure 1 viruses-13-02477-f001:**
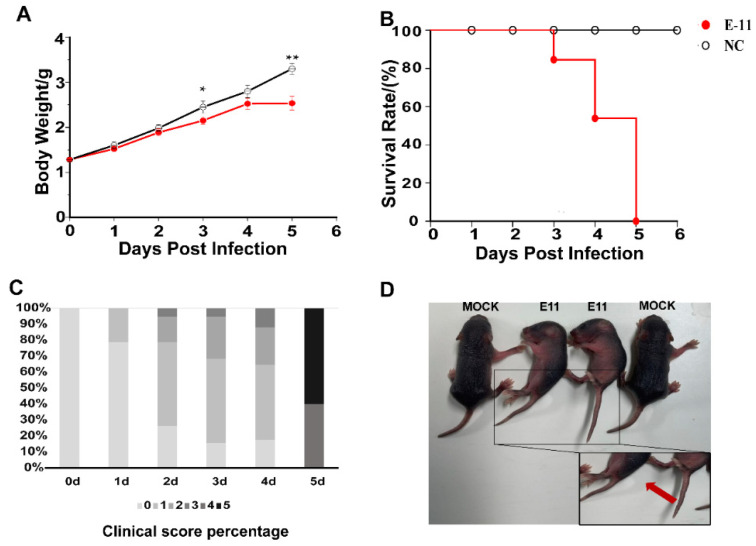
Body weights, survival rates of mice, clinical score percentage, and symptoms in the experiments. Two-day-old *IFNAR*−/− mice (n = 16 per group) were inoculated intracranially with 40 µL of E11 suspension (virus titre, 10^8^ TCID50/mL, diluted 10-fold to 10^7^ TCID50/mL). Control animals were administered culture medium instead of virus. Body weight (**A**), survival rates (**B**), clinical score percentage (**C**), and clinical symptoms (**D**) were monitored and recorded daily after inoculation with E11 until 6 dpi. Data represent the mean results of 16 mice ± standard error of the mean; * *p* < 0.05; ** *p* < 0.01; NC, negative control.

**Figure 2 viruses-13-02477-f002:**
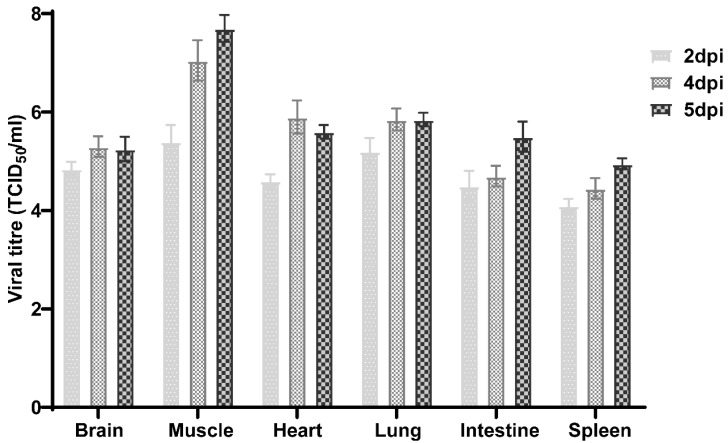
Changes in E11 viral load in different tissues at 2, 4, and 5 dpi. Two-day-old neonatal *IFNAR*−/− mice were challenged with 40 µL of E11 strain HuB2019-6(virus titre, 10^8^ TCID50/mL, diluted 10-fold to 10^7^ TCID50/mL) and the viral loads in the brain, hind limb muscle, heart, lung, intestine, and spleen tissues were determined by TCID_50_. Data represent the mean results of three mice ± standard error of the mean with five biological replicates.

**Figure 3 viruses-13-02477-f003:**
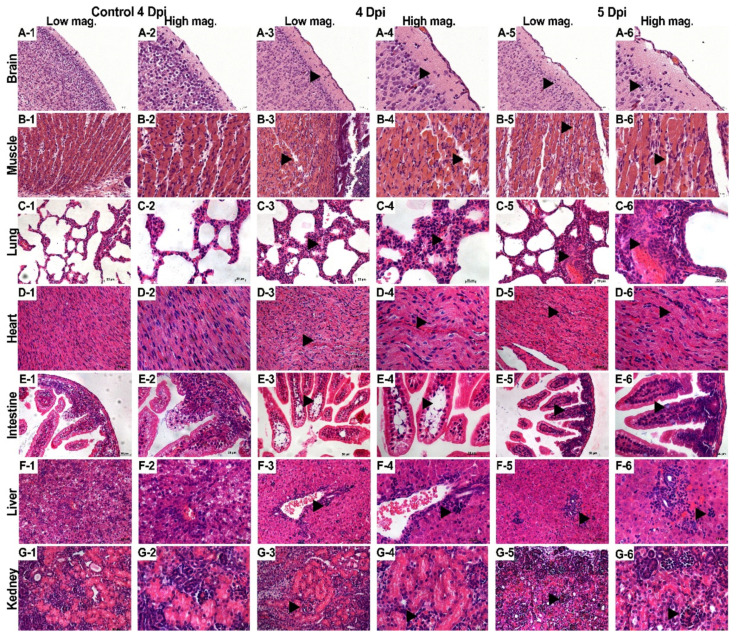
Histopathological examination. After the two-day-old neonatal *IFNAR*−/− mice were challenged intracranially with 40 µL of E11 suspension (virus titre, 10^8^ TCID50/mL, diluted 10-fold to 10^7^ TCID50/mL), tissue samples from the brain (**A** 1 to 6), hind limb muscle (**B** 1 to 6), heart (**C** 1 to 6), lung (**D** 1 to 6), and intestine (**E** 1 to 6) were subjected to pathological analysis. No obvious histological changes (1, 2) were observed in each tissue of the negative control group. Mice with grade 4 or 5 clinical symptoms infected with E11(HuB2019-16) exhibited cerebral oedema (**A**); massive damage was observed in the limb muscles, characterised by the foci of myositis and myonecrosis associated with neutrophilic infiltration (**B**); alveolar shrinkage, pulmonary fibrosis, and viral pneumonia in the lung tissue (**C**); severe myocardial tears and capillary leakage in the cardiac tissue (**D**); and vacuolar degeneration of small intestinal villi accompanied by massive inflammatory cell infiltrates (**E**); viral hepatitis is characterized by hepatocellular edema with lymphocytic infiltration (**F**); nephritis is characterized by glomerulocytosis (**G**). Observations of singular columns (1, 3, 5) were made at a magnification of 20×. The other panels were observed at 40×. All experiments were repeated three times.

**Figure 4 viruses-13-02477-f004:**
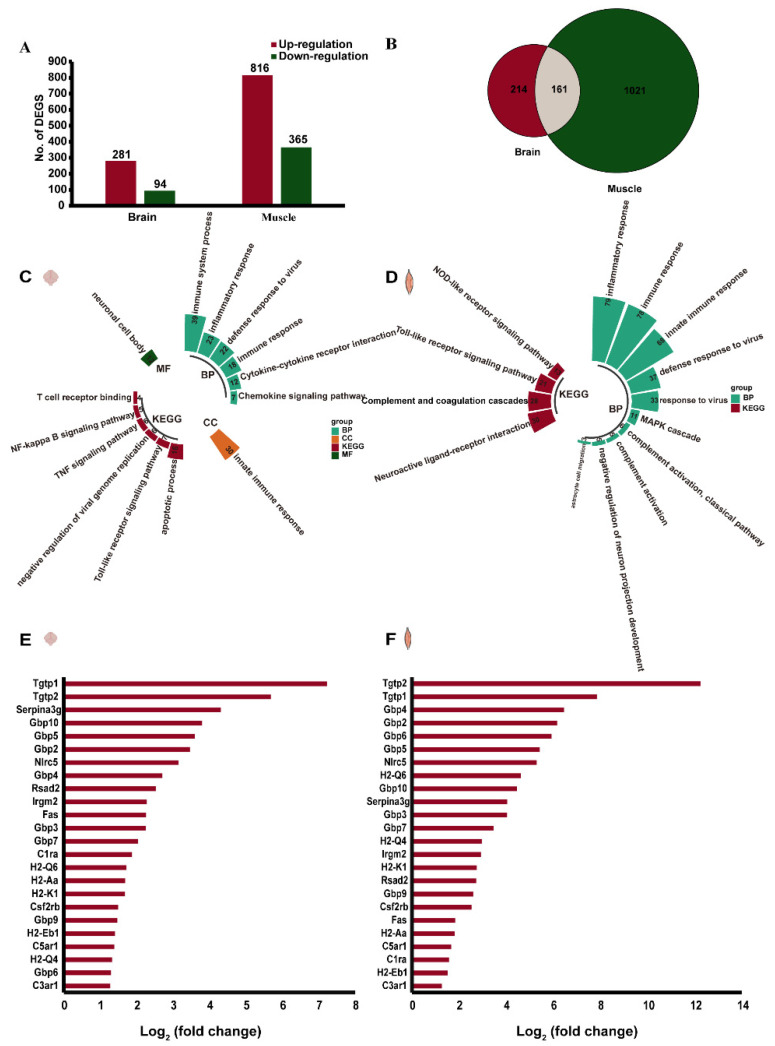
Visualisation of gene expression changes and their biological pathways. Transcriptome analysis of the brain and skeletal muscle tissues from 2-day-old mice infected with 10^7^ TCID50 via intracranial injection at 4 dpi. The detailed number of upregulated and downregulated differentially expressed genes (DEGs) in each group (**A**). Venn diagram displaying the number of brain-specific, muscle-specific, and brain/muscle-common DEGs after E11 inoculation (**B**). GO functional analysis and KEGG pathways of differentially expressed genes in the brain and skeletal muscle (**C**,**D**). Visualisation of DEGs involved in antiviral and other related immune response pathways (**E**,**F**).

**Figure 5 viruses-13-02477-f005:**
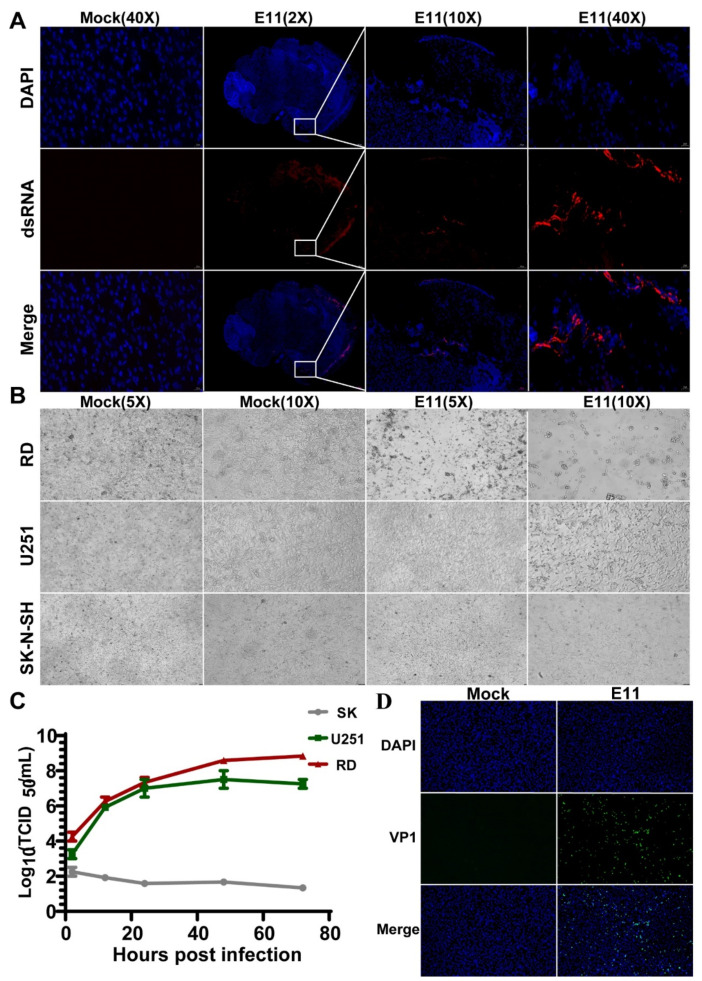
Virus replication in mouse brain tissues and neural cells. (**A**) Brain tissue samples of 2-day-old mice challenged with 10^7^ TCID50 via intracranial injection, sacrificed at 4 dpi, stained with dsRNA against E11 (red), and counterstained with DAPI (blue) for detection of nuclei. Scale bar = 50 μm. (**B**) U251 and SK-N-SH cells were infected with E11 at an MOI of 10 as described above. Cells were visualised at 24 hpi under bright-field microscopy at 5/10× magnification. (**C**) Infection and replication of E11 virus in U251 and RD cells infected with MOI = 10/0.005, respectively. Infection medium was removed at 2 hpi to reduce background. Cell culture lysates were collected at four time points (12 hpi, 24 hpi, 48 hpi, and 72 hpi) after infection, and viral titres were measured using endpoint dilutions for growth in RD cells. Error bars represent standard error of the mean (SEM) from three biological replicates. (**D**) U251 cells were infected with E11 at an MOI of 10. Cells were fixed at 24 hpi and stained with FITC (green) against E11 and counterstained with DAPI (blue) for the detection of nuclei.

## Data Availability

Not applicable.

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
