# Peer review of "Pathological Features of Echovirus-11-Associated Brain Damage in Mice Based on RNA-Seq Analysis"

_viruses, 2021, doi:10.3390/v13122477_

Round 1

Reviewer 1 Report

The manuscript titled: “Pathological Features of Echovirus 11-Associated Brain Damage in Mice Based on RNA-Seq Analysis” (viruses-1492360);

The article covers an important topic associated with pathogenicity of enteroviruses, viruses that can be associated with the emergence of life-threatening neurological disease.

The interaction between pathogen and the host immune system is critical to develop and understanding of the mechanisms of viral pathogenesis.

Overall, the content of the manuscript is well-organized and well-written. However, as I read the article, I came upon a few things which were not clear or obvious to the reader. There are also a few minor errors.

They are summarized as follows:

Line 34: Please take into account that purpura is a kind of skin rash and a rash is a symptom, not a disease;

Line 50: It is “Am” – it should be “AM”.

Methods, 2.4 Virus titration: Please clarify protocol of the study; It is unclear how many mice were used in the monitoring of viral loads in different organs. Did you perform virus titration from equivalent weights of tissue samples? The monitoring of virus replication in cell cultures was not described in “Methods” (Fig. 5C – The growth curve for E11).

Line 158: It is “VP1 of Echovirus”; It should be “VP1 of Echovirus 11”;

Line 170: “The same” in this sentence means that the clinical symptoms in people and mice were identical. Are you sure it is justified?

Line 186-187: The sentence: “The viral load in the brain was the highest” is confusing and is not supported by the Figure 2.

Line 204-206: an unfinished sentence;

Line 250-251: a calculation error (284+94=378 not 375);

Line 264-267: This sentence is confusing. You twice listed “complement activation” and “response to a virus”. If they are different categories of genes, you should make this clear.

Line 294: Immunofluorescence staining revealed dsDNA, not E11, it should be mentioned.

Line 298-299: You used MOI 10 for U251 and SK-N-SH and MOI 0,005 for RD cells, the figure 5C contradicts this (see the first point on the curve);

Figure 5C: It is incomprehensible the lack of a curve for E11 replication in SK-N-SH cells in Figure 5C. The absence of a cytopathic effect at 24h post-infection does not necessarily mean that the virus doesn’t replicate.

Line 380: Typo error: It is “rela ted”; It should be “related”.

Author Response

Reviewer 1:

The article covers an important topic associated with pathogenicity of enteroviruses, viruses that can be associated with the emergence of life-threatening neurological disease.

The interaction between pathogen and the host immune system is critical to develop and understanding of the mechanisms of viral pathogenesis.

Overall, the content of the manuscript is well-organized and well-written. However, as I read the article, I came upon a few things which were not clear or obvious to the reader. There are also a few minor errors.

Response to reviewer 1 comment 1: Thank you for the positive evaluation, we revised the manuscript carefully according to your comments.

Line 34: Please take into account that purpura is a kind of skin rash and a rash is a symptom, not a disease;

Response to reviewer 1 comment 2:Thanks for the comments, we agreed with you and deleted purpura and rash. (Page 1, line 34).

Line 50: It is “Am” – it should be “AM”.

Response to reviewer 1 comment 3: Sorry for the mistake, revised as suggested, term “Am” was changed to “AM”. (Page 2, line 49).

Methods, 2.4 Virus titration: Please clarify protocol of the study; It is unclear how many mice were used in the monitoring of viral loads in different organs. Did you perform virus titration from equivalent weights of tissue samples? The monitoring of virus replication in cell cultures was not described in “Methods” (Fig. 5C – The growth curve for E11).

Response to reviewer 1 comment 4: Sorry for the ambiguous expression. It has been clarified in the manuscript. The viral titers from 10-fold serial dilutions of each sample were evaluated using a 50% TCID50assay on RD cells. Repeat the measurement 3 times at each time point (n=5). U251 and SK-N-SH cells were infected with a multiplicity of infection (MOI) at 10, and RD cell was infected with a MOI at 0.005. Infection medium was washed three times with PBS at 2 hours post infection (hpi) to reduce background. Cell culture lysates were collected at four time points (12 hpi, 24 hpi, 48 hpi, and 72 hpi) after infection, and the viral titres were measured using endpoint dilutions for growth in RD cells. Repeat the measurement 3 times at each time point. (Page 3, lines 120-127).

Line 158: It is “VP1 of Echovirus”; It should be “VP1 of Echovirus 11”;

Response to reviewer 1 comment 5: Sorry for the mistake, revised as suggested. Term “VP1 of Echovirus” was revised as “VP1 of Echovirus 11” VP1 of Echovirus 11 (Reacts with the Echovirus 4, 6, 9, 11, 30 and 34). (Page 4, line 166).

Line 170: “The same” in this sentence means that the clinical symptoms in people and mice were identical. Are you sure it is justified?

Response to reviewer 1 comment 6: It has been clarified in the manuscript. We have amended the expression as followed: Further, the clinical symptoms produced by the inoculation of 10-fold TCID50-diluted E11 in mice that can simulate the symptoms of human infant infections. (Page 4, lines 176-178).

Line 186-187: The sentence: “The viral load in the brain was the highest” is confusing and is not supported by the Figure 2.

Response to reviewer 1 comment 7: Sorry for the mistake, and it has been clarified in the manuscript. It should be “The viral load in the muscle was the highest”. (Page 5, line 195).

Line 204-206: an unfinished sentence;

Response to reviewer 1 comment 8: Sorry for the mistake, and it has been clarified in the manuscript. 2-day-old neonatal mice were challenged with the E11 HuB2019-16 strain (virus titre, 108 TCID50/ml, diluted 10-fold to 107 TCID50/ml) via the i.c. route and animals were sacrifice when the clinical scores reached 4; the tissues were then surgically removed, processed for histological sectioning, and subjected to HE staining. (Page 6, lines 213-217).

Line 250-251: a calculation error (284+94=378 not 375);

Response to reviewer 1 comment 9: Sorry for the calculation error, and it has been clarified in the manuscript. we wrote the number 284 wrong and it should be 281. We have corrected it. (Page 7, lines 261-262).

Line 264-267: This sentence is confusing. You twice listed “complement activation” and “response to a virus”. If they are different categories of genes, you should make this clear.

Response to reviewer 1 comment 10: Thank you for your suggestion, and it has been clarified in the manuscript. In the muscle, the differential expression of genes involved in inflammatory response, immune response, innate immune response, defense response to a virus, response to a virus, MAPK cascade, complement activation (classical pathway), complement activation, negative regulation of neuron projection development, and astrocyte cell migration was found. (Page 8, lines 276-279).

Line 294: Immunofluorescence staining revealed dsDNA, not E11, it should be mentioned.

Response to reviewer 1 comment 11: Sorry for the mistake and revised as suggested. Immunofluorescence staining revealed that the viral dsRNA was detected in the mouse brain region. (Page 9, lines 304-305).

Line 298-299: You used MOI 10 for U251 and SK-N-SH and MOI 0,005 for RD cells, the figure 5C contradicts this (see the first point on the curve);

Response to reviewer 1 comment 12: It has been clarified in the manuscript and Figure 5. The first point on the curve was 2 hpi with E11 infection. We use E11 to infect neural cells and RD cells with MOI = 10/0.005, respectively. The three cells had different sensitivities to the virus. RD cell is a sensitive cell line of enteroviruses, and its binding efficiency with E11 is much higher than that of the other two cells. Therefore, the number of viruses adsorbed to the three cells was different. When the virus has not completed the first round of replication, we washed it three times with PBS after 2 hours of infection for virus titration. Therefore, the viral titres must be inconsistent with MOI.

Figure 5C: It is incomprehensible the lack of a curve for E11 replication in SK-N-SH cells in Figure 5C. The absence of a cytopathic effect at 24h post-infection does not necessarily mean that the virus doesn’t replicate.

Response to reviewer 1 comment 13: Thanks for the comments, we agreed with you. Actually, we did this experiment, but we didn’t show the results. Now, we add this result (Figture 5C).

Line 380: Typo error: It is “rela ted”; It should be “related”.

Response to reviewer 1 comment 14: Sorry for the mistake and revised as suggested. we have modified “rela ted” to “related”. (Page 12, line 413)

Reviewer 2 Report

The manuscript submitted by Guoyan Zhang (viruses-1492360) focuses on the neuropathology and neurovirulence of E11 infections, using a model of E11 infection in 2-day-old interferon receptor knockout (IFNAR-/-) mice, which are susceptible to enteroviruses. In this study are reported the pathological changes of several E11 infected mice tissues (brain, hindlimb muscles, heart, lung and small intensine tissues) for simulating the clinical symptoms of E11 infection in neonatal children. At a molecular level a transcriptome sequence analysis of the collected  brain and hindlimb muscles tissues was conducted in order the gene expression profile and the changes of the host genes caused by E11 infections.

This is an interesting study as elucidates the neuropathology and neurovirulence of fatal E11 strain causing aseptic meningitides in neonatal and the changes responed to E11 infection at a molecular level.  

This manuscript is very well written and the research was done using a straightforward methodology and achieved interesting results to the field, as our knowledge about the pathogenic mechanism of E11 infection is limited.

Below are listed my specific comment for the manuscript.

  1. page 3, line102: please explain why intracranially injection was selected for the mice infection.
  2. Page 3, lines 102-103: the infection dose of E11 suspension was 40 μl of 108 TCID50/ml (4x106 TCID50/mouse) or 40 μl of 107TCID50/ml (4x105 TCID50/mouse) ?
  3. Page 3, line 113: please explain the abbreviation d.p.i for the days post infection
  4. Page 4, line 168: see comment 2
  5. Page 5, line 176: see comment 2
  6. Page 5, lines 185-186: The sentence “The results showed that he viral…..at 2 d.p.i (figure 2)” is unclear.
  7. Page 5, line 186-187: The sentence “The viral load in the brain was the highest” has to be reformed as this is not obvious in Figure 2.
  8. Page5, line 188: the phrase “..in the middle and late stages of injection.”  Has to be corrected as ““..in the middle  and late stages of infection.”   
  9. Page 5, line197 and line 205: see comment 2
  10. Page 7, line 235: see comment 2
  11. Page7, section 3.4: please explain under what criteria the tissues of brain and muscles were selected for transcriptional analysis?

Author Response

Reviewer 2:

The manuscript submitted by Guoyan Zhang (viruses-1492360) focuses on the neuropathology and neurovirulence of E11 infections, using a model of E11 infection in 2-day-old interferon receptor knockout (IFNAR-/-) mice, which are susceptible to enteroviruses. In this study are reported the pathological changes of several E11 infected mice tissues (brain, hindlimb muscles, heart, lung and small intensine tissues) for simulating the clinical symptoms of E11 infection in neonatal children. At a molecular level a transcriptome sequence analysis of the collected brain and hindlimb muscles tissues was conducted in order the gene expression profile and the changes of the host genes caused by E11 infections.

This is an interesting study as elucidates the neuropathology and neurovirulence of fatal E11 strain causing aseptic meningitides in neonatal and the changes responed to E11 infection at a molecular level.  

This manuscript is very well written and the research was done using a straightforward methodology and achieved interesting results to the field, as our knowledge about the pathogenic mechanism of E11 infection is limited.

Response to reviewer 2 comment 1: thank you very much for the positive comments, we revised the manuscript carefully according to your comments.

Below are listed my specific comment for the manuscript.

1, page 3, line102: please explain why intracranially injection was selected for the mice infection.

Response to reviewer 2 comment 2: It has been clarified in the manuscript. Intracranial injection, intraperitoneal injection and intramuscular injection are the three most commonly used injection methods at present. Through the pilot experiment, we found that the symptoms of mice were most obvious by intracranially injection, and intracranial injection can better simulate the clinical symptoms of infant infection. Therefore, we selected intracranial injection as the experimental method. (Page 3, lines 103-105).

2, Page 3, lines 102-103: the infection dose of E11 suspension was 40 μl of 108 TCID50/ml (4x106 TCID50/mouse) or 40 μl of 107TCID50/ml (4x105 TCID50/mouse)?

Response to reviewer 2 comment 3: Sorry for the mistakes, and it has been clarified in the manuscript. 40 µL of the E11 suspension (virus titre, 108 TCID50/ml, diluted 10-fold to 107 TCID50/ml). (Page 3, lines 104-105).

3, Page 3, line 113: please explain the abbreviation d.p.i for the days post infection

Response to reviewer 2 comment 4: Thank you for the comments and revised as suggested. (Page 3, line 115).

4, Page 4, line 168: see comment 2

Response to reviewer 2 comment 5: Sorry for the mistakes, we modified the expression as followed the infection dose of E11 suspension was 40 μl of 108 TCID50/ml or 40 μl of 107TCID50/ml. (Page 4, lines 176-177).

5, Page 5, line 176: see comment 2

Response to reviewer 2 comment 6: Sorry for the mistakes, and it has been clarified in the manuscript. 40 µL of the E11 suspension (virus titre, 108 TCID50/ml, diluted 10-fold to 107 TCID50/ml). (Page 5, lines 185).

6, Page 5, lines 185-186: The sentence “The results showed that the viral…..at 2 d.p.i (figure 2)” is unclear.

Response to reviewer 2 comment 7: Thank you for the comments, and it has been clarified in the manuscript. The results showed that the viral loads differed significantly among tissues (Figure 2). And the viral loads in each organ were low at 2 dpi (Figure 2). The viral load in the muscle was the highest. (Page 5, lines 210-212).

7, Page 5, line 186-187: The sentence “The viral load in the brain was the highest” has to be reformed as this is not obvious in Figure 2.

Response to reviewer 2 comment 8: Thank you for pointing out our mistake, we modified “brain” to “muscle”. The viral load in the muscle was the highest. (Page 5, line 195).

8, Page5, line 188: the phrase “..in the middle and late stages of injection.”  Has to be corrected as ““..in the middle  and late stages of infection.”   

Response to reviewer 2 comment 9: Revised as suggested. Term “injection” was revised as “infection”. (Page 5, lines 196-197).

9, Page 5, line197 and line 205: see comment 2

Response to reviewer 2 comment 10: Sorry for the mistakes, and it has been clarified in the manuscript. 40 µL of the E11 suspension (virus titre, 108 TCID50/ml, diluted 10-fold to 107 TCID50/ml). (Page 6, lines 214-215 and lines 206-207).

10, Page 7, line 235: see comment 2

Response to reviewer 2 comment 11: Sorry for the mistakes, and it has been clarified in the manuscript. 40 µL of the E11 suspension (virus titre, 108 TCID50/ml, diluted 10-fold to 107 TCID50/ml). (Page 7, lines 245-246).

11, Page7, section 3.4: please explain under what criteria the tissues of brain and muscles were selected for transcriptional analysis?

Response to reviewer 2 comment 12: It has been clarified in the manuscript. Muscle is the most effective tissue for enterovirus replication, and E11 used in this study came from a case of neonatal aseptic meningitis. Therefore, our study focused on the brain and muscle with the highest replication efficiency.
